

# Effect of alpha-linolenic acid on aminoglycoside nephrotoxicity and RhoA/Rho-kinase pathway in kidney

Percin Pazarci[1], Serkan Özler[2] and Halil Mahir Kaplan[3]

[1] Department of Medical Biology, Cukurova University Faculty of Medicine, Adana, Turkey
[2] Department of Urology, Mustafa Kemal University Faculty of Medicine, Hatay, Turkey
[3] Department of Pharmacology, Cukurova University Faculty of Medicine, Adana, Turkey

## ABSTRACT

Aminoglycoside nephrotoxicity stands as a primary contributor to the development of acute intrinsic renal failure. Distinctive characteristic associated with this nephrotoxicity is the occurrence of tubular necrosis, which is why it is commonly referred to as acute tubular necrosis. Studies have demonstrated that inhibiting rhoA/rho-kinase pathway is beneficial for kidney damage induced by diabetes and renal ischemia. Comparable pathological conditions can be observed in aminoglycoside nephrotoxicity, like those found in diabetes and renal ischemia. Gentamicin, an aminoglycoside, is known to activate Rho/Rho-kinase pathway. The primary goal of this study is to explore influence of oxidative stress on this pathway by concurrently administering gentamicin and alpha-linolenic acid (ALA) possessing known antioxidant properties. To achieve this, gentamicin ($100$ mg kg$^{-1}$) and ALA ($70$ mg kg$^{-1}$) were administered to mice for a period of 9 days, and Rho/Rho-kinase pathway was examined by using ELISA. Administration of gentamicin to mice led to an elevation in RhoA and rho-kinase II levels, along with the activity of rho-kinase in kidneys. However, ALA effectively reversed this heightened response. ALA, known for its antioxidant properties, inhibited activation of Rho/Rho-kinase pathway induced by gentamicin. This finding suggests that gentamicin induces nephrotoxicity through oxidative stress.

# INTRODUCTION

Rho proteins are monomeric GTPases that belong to the Ras superfamily's Rho subfamily. RhoA, RhoB, and RhoC have comparable biological roles and share an amino acid sequence in their effector regions. Studies involving RhoA served as the foundation for many of the Rho's reported functions (*Fukata, Amano & Kaibuchi, 2001*). The subtype of Rho protein that is most prevalent and well-studied in the body is RhoA. Numerous receptors trigger the RhoA protein, which in turn activates the rho-kinase enzyme (*Boettner & Van Aelst, 2002*; *Miao et al., 2002*).

Rho kinase, aptly named for its central role in transmitting RhoA's commands, serves as a molecular translator, converting biochemical signals into intricate cellular responses. This

Corresponding author
Percin Pazarci,
percinpazarci@gmail.com

dynamic duo has been implicated in processes as diverse as cytoskeletal rearrangement, cell adhesion, smooth muscle contraction, and cell migration (*Guan et al., 2023*).

ROCK1 and ROCK2 are isoform protein serine/threonine kinase enzymes activated by RhoA (*Fukata, Amano & Kaibuchi, 2001*; *Kimura et al., 1996*). In humans, the ROCK1 gene is situated on chromosome 18 (18q11.1), while the ROCK2 gene resides on chromosome 2 (2p24) (*Chitaley & Webb, 2002*). Studies have indicated that while ROCK2 shows higher expression levels in the brain and heart, ROCK1 is more prominently expressed in the testis, lung, kidney, with the Rho-kinase enzyme presence confirmed in nearly all tissues (*Buyukafsar & Levent, 2003*; *Buyukafsar, Levent & Ark, 2003*; *Buyukafsar & Un, 2003*; *Fukata, Amano & Kaibuchi, 2001*; *Kimura et al., 1996*; *Miao, Dai & Zhang, 2002*). The roles of ROCK1 and ROCK2 in tubular necrosis are multifaceted. These kinases are pivotal in cytoskeletal reorganization, which is essential for maintaining the integrity of tubular epithelial cells (*Shi et al., 2013*). Disruption in cytoskeletal dynamics can lead to cell injury and death, contributing to tubular necrosis (*Molitoris, 2004*). Additionally, ROCK1 and ROCK2 are involved in mediating inflammatory responses. Studies have shown that inhibition of these kinases can reduce inflammation and improve kidney function in models of acute renal injury (*Kentrup et al., 2011*).

Previous research has demonstrated that the Rho/Rho-kinase (RRK) pathway is activated in diabetes and renal ischemia, and inhibition of this pathway reduces nephron damage (*Komers, 2013*; *Prakash et al., 2008*). Pathological conditions caused by the increase in intracellular superoxide radicals are observed in diabetes and renal ischemia, as seen in aminoglycoside nephrotoxicity (*DeRubertis, Craven & Melhem, 2007*; *Yin et al., 2001*).

Gentamicin, an aminoglycoside, can cause nephrotoxicity leading to acute tubular necrosis and renal failure by accumulating in the renal proximal convoluted tubules. This accumulation triggers oxidative stress, generating reactive oxygen species (ROS) that damage cellular components (*Quiros et al., 2011*). Additionally, gentamicin induces endoplasmic reticulum stress and apoptosis, stimulates pro-inflammatory cytokine production, causes phospholipidosis by disrupting lysosomal function, and disrupts calcium homeostasis (*Abouzed et al., 2021*; *McWilliam et al., 2017*). These mechanisms collectively contribute to tubular cell injury and necrosis, highlighting the importance of understanding these pathways to develop strategies to mitigate gentamicin-induced nephrotoxicity. Gentamicin renowned for its bactericidal prowess, has long been utilized to combat life-threatening infections caused by a range of Gram negative/positive bacteria. However, recent research has unveiled gentamicin's potential to influence cellular processes beyond its primary antimicrobial role. Gentamicin is known to increase the expression and activity of RhoA and Rho-kinase enzymes in nephrons (*Kaplan, 2016*).

Alpha-linolenic acid (ALA) is a potent antioxidant that helps reduce oxidative stress by neutralizing ROS and enhancing the body's antioxidant defenses. ALA can regenerate other antioxidants, such as glutathione, and improve mitochondrial function, thereby reducing the production of ROS (*Alam et al., 2021*). By mitigating oxidative stress, ALA helps protect cells from damage, inflammation, and apoptosis, which are key factors in conditions like gentamicin-induced nephrotoxicity (*Yan et al., 2024*). This study aimed to

investigate the effect of oxidative stress on this pathway by applying gentamicin together with ALA, known for its antioxidant properties.

## MATERIALS & METHODS

### Animals

This study employed male, albino, Balb/c mice aged 8 weeks, which were acquired from the Çukurova University Experimental Animal Centre. Ethical clearance for this research was obtained from the Cukurova University Animal Care and Ethics Committee (protocol code 2016-2-13).

The mice were housed in ventilated cages (21 °C, 12-hour dark light cycle) within a pathogen-free animal facility, with free access to food and water. The mice were categorized randomly into three distinct groups: control, gentamicin and ALA groups (n = 8 for each group, total 24 mice). In the gentamicin group, gentamicin (Catalog#: GEN-10B; Capricorn Scientific, Ebsdorfergrund, Germany), was administered intraperitoneally ($100 \, \text{mg} \, \text{kg}^{-1}$) for a duration of 9 days. The ALA group received intraperitoneal gentamicin ($100 \, \text{mg} \, \text{kg}^{-1}$) and gavage-administered ALA ($70 \, \text{mg} \, \text{kg}^{-1}$) (CAS: 463-40-1; Sigma-Aldrich, St. Louis, MO, USA) for 9 days. The control group received intraperitoneal physiological serum for 9 days under identical conditions. Following the outlined protocol, cervical dislocation was carried out on the mice (*Kaplan et al., 2016a*; *Kaplan et al., 2016b*). The mice's kidneys were dislocated after cervical dislocation and preserved at $-80 \, °C$ for subsequent use. Levels of rhoA and rho kinase, along with the rho kinase enzyme activity were measured from the obtained samples. No anesthesia or analgesia was administered during any procedures, in accordance with the protocol and bioethics approval. Mice were to be euthanized before the planned end of the experiment if they showed severe signs of distress, but no such incidents occurred.

### Homogenization of tissues

Samples of frozen tissue are treated with a mixture consisting of 3 mL per gram of tissue of radio-immunoprecipitation assay buffer (Product#: R0278; Sigma-Aldrich, St. Louis, MO, USA). This solution is augmented with the inclusion of sodium vanadate (Product#: 5.08605; SigmaAldrich, St. Louis, MO, USA), phenylmethanesulfonylfluoride (Product#: P7626; Sigma-Aldrich, St. Louis, MO, USA) and protease inhibitor (Product#: P8340; Sigma-Aldrich, St. Louis, MO, USA) (30 µL of each). The homogenization of these tubes is achieved through ultrasonication, while they are maintained on ice. Following homogenization, the resulting mixtures are centrifuged (Eppendorf 5425) to collect supernatants (10,000 rpm/10 minutes).

### Quantification of proteins

The protein content of the homogenized tissues is measured by Bradford method. To establish a protein standard curve, solutions with known concentrations are prepared using bovine serum albumin (Product#: B8667; Sigma-Aldrich, St. Louis, MO, USA) at concentrations of 1, 2, 4, 6, 8, and 10 µg/mL. Afterwards, a volume of 10 µL is extracted from each sample and subsequently diluted to reach a final volume of 100 µL using distilled

water. Finally, the prepared standards and diluted samples are provided with Bradford solution (Product#: 1.15444; Sigma-Aldrich, St. Louis, MO, USA) (1ml). The resulting mixtures are vortexed to ensure thorough mixing, and the absorbance at 595 nm is manually measured. By comparing the absorbance results to the standard curve produced by the Prism software, protein quantification is accomplished.

## ELISA experiments

The ELISA is conducted following the manufacturer's (Mybiosource, San Diego, CA) instructions to assess the levels of RhoA (Catalog#: MBS2516172) and Rho kinase II (Catalog#: MBS3807341), along with the determination of Rho kinase enzyme activity (Catalog#: MBS168354).

According to manufacturer's instructions, first, 100 μL of the standard or sample is added to each well and incubated for 90 minutes at 37 °C. After incubation, the liquid is removed, and 100 μL of Biotinylated Detection Antibody is added, followed by an hour of incubation at 37 °C. The wells are then aspirated and washed three times. Next, 100 μL of HRP Conjugate is added and incubated for 30 minutes at 37 °C. The wells are aspirated and washed five times. Then, 90 μL of Substrate Reagent is added and incubated for 15 minutes at 37 °C. Finally, 50 μL of Stop Solution is added, and the results are read at 450 nm by using Biochrom EZ Read 400 immediately.

## Statistical analysis

The results are reported as means $\pm$ SEM. To assess differences in the results across groups, a statistical analysis is conducted using one-way analysis of variance (one-way ANOVA), with adjustments made for multiple comparisons *via* the Bonferroni correction method. Statistical significance is defined as *p*-values less than 0.05.

# RESULTS

Although gentamicin administration in mice caused an upregulation of Rho-A (Fig. 1) and Rho-kinase II (Fig. 2), ALA reversed this increase. Furthermore, gentamicin administration also led to an elevation in Rho-kinase activity, and ALA effectively reversed this escalation (Fig. 3). Mean, S.E.M. and p values of Rho-A, Rho-kinase II concentrations and Rho-kinase activity for control, gentamicin and gentamicin+ALA treated groups are given in Table 1.

# DISCUSSION

Oxidative stress, the imbalance between free radicals and the body's ability to neutralize them, lies at the core of numerous health maladies, including aging, cardiovascular diseases, neurodegenerative disorders, and even nephron damage (*Cachofeiro et al., 2008*; *Yang et al., 2024*). ALA, known for its antioxidant properties, is a potential guardian against this relentless assault on our cellular landscape.

Gentamicin is a widely used aminoglycoside antibiotic that causes nephrotoxicity by several mechanisms, including oxidative stress, inflammation, apoptosis, and phospholipidosis (*Quiros et al., 2011*). This study investigated the effects of the chronic application of gentamicin on the RRK pathway, which represents one of the essential

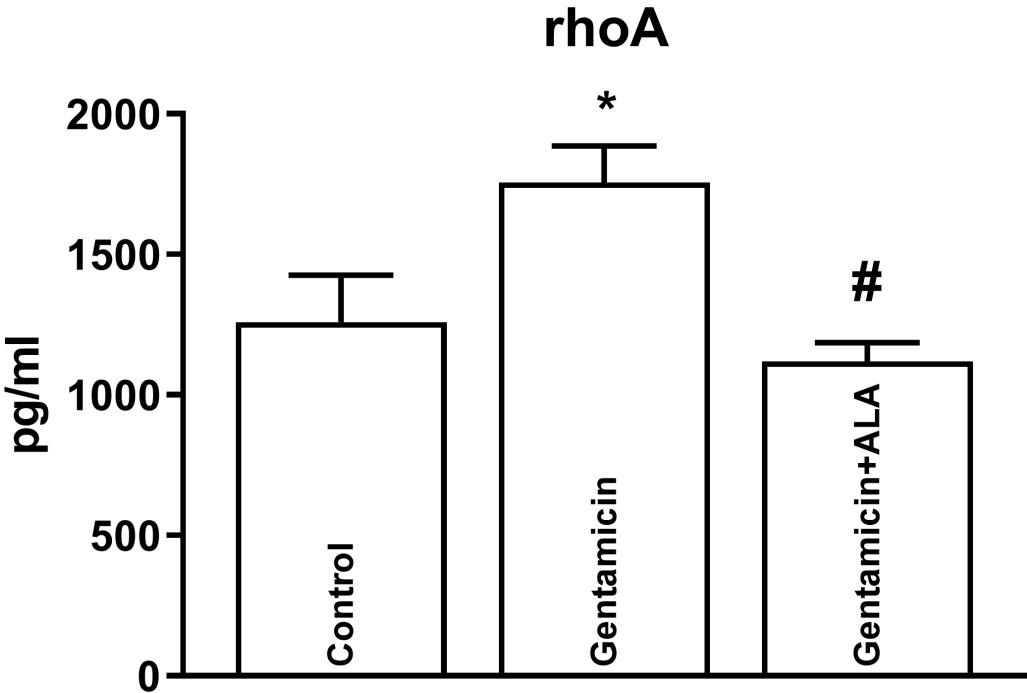

**Figure 1** **The impact of gentamicin and ALA treatments on RhoA (measured by ELISA) in the kidneys.**
*: $p < 0.05$ when compared to control, #: $p < 0.05$ when compared to gentamicin treated group, $n = 8$.

intracellular signaling pathways that plays a vital role in cell proliferation and development. RhoA performs various functions after G protein-coupled receptors are activated by specific agonists (*DeRubertis, Craven & Melhem, 2007*). This protein activates the Rho-kinase enzyme, which is involved in many intracellular activities (*Buyukafsar & Levent, 2003*; *Buyukafsar, Levent & Ark, 2003*; *Buyukafsar & Un, 2003*). The RRK pathway has been linked to several kidney diseases, including hypertensive glomerulosclerosis, interstitial fibrosis, and diabetic nephropathy (*Jiang et al., 2016*; *Ruperez et al., 2005*). However, it is still unclear how oxidative stress affects the activation of the RRK pathway by gentamicin.

In this study, gentamicin administration upregulated the RhoA and the Rho-kinase. It also elevated the activity of the Rho kinase. Several studies indicate that reactive oxygen products, which increase due to oxidative stress, activate the RRK pathway (*Jin, Ying & Webb, 2004*; *Kajimoto et al., 2007*; *MacKay et al., 2017*). The mechanism by which oxidative stress activates the RRK pathway in gentamicin-induced renal injury is not fully understood. Oxidative stress induces the formation of ROS. Possibly, ROS can directly activate RhoA by oxidizing its cysteine residues (*Aghajanian et al., 2009*). Alternatively, oxidative stress might induce the activation of angiotensin II, a potent stimulator of the RRK pathway. Angiotensin II can bind to its type 1 receptor and activate RhoA *via* G-protein-coupled receptor signaling. Angiotensin II can also stimulate the production of ROS, further amplifying the activation of RhoA (*Banday & Lokhandwala, 2011*).

The activation of the RRK pathway, as demonstrated in our study, might be due to oxidative stress induced by chronic gentamicin administration in nephrons. Gentamicin

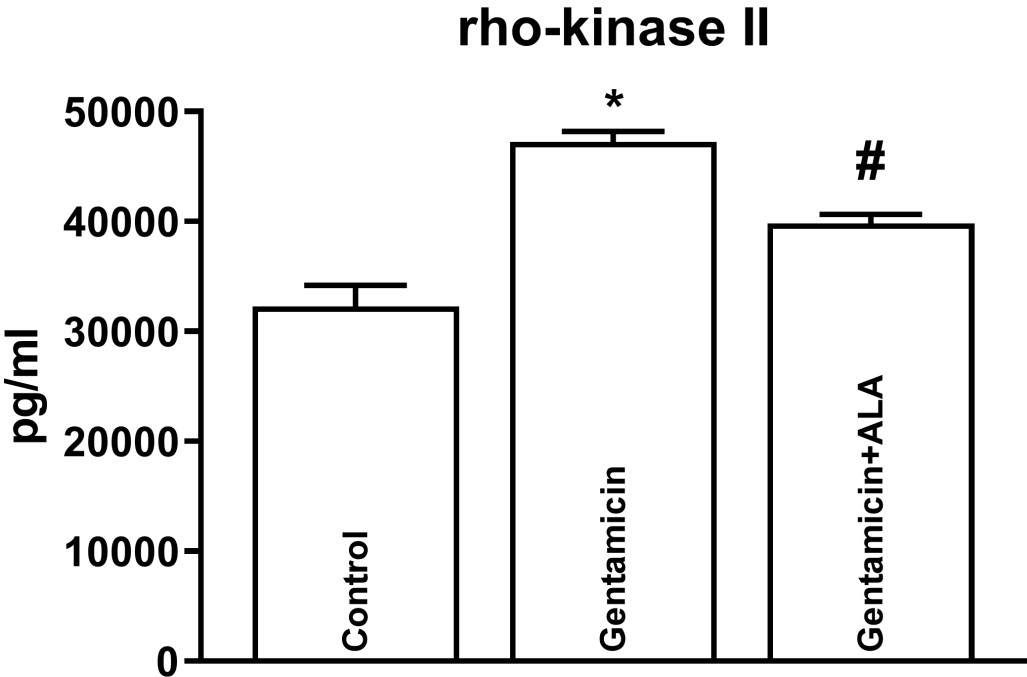

**Figure 2   The impact of gentamicin and ALA treatments on Rho-kinase II (measured by ELISA) in the kidneys.** *: $p < 0.05$ when compared to control, #: $p < 0.05$ when compared to gentamicin treated group, $n = 8$.

has been shown to elevate intracellular calcium levels while inducing oxidative stress (*Malis & Bonventre, 1986*; *Rordorf, Koroshetz & Bonventre, 1991*; *Verity, 1993*). Additionally, the COX-2 enzyme, an inflammatory mediator, and the prostaglandins it produces contribute to gentamicin-induced nephrotoxicity. Evidence shows that gentamicin administration increases COX-2 and phospholipase A2 enzymes in nephrons (*Raso et al., 2002*; *Rordorf, Koroshetz & Bonventre, 1991*; *Verity, 1993*). Research has indicated that gentamicin induced nephron damage is linked to the iNOS (*Lee et al., 2013*). The increased activity of the RRK signaling pathway might be linked to the activation of these enzymes. Further studies are needed to confirm this.

This study also demonstrates the ability of ALA to reverse the gentamicin-induced upregulation of Rho-A protein and Rho-kinase II enzyme in kidneys. Furthermore, ALA effectively counteract the rise in Rho-kinase enzyme activity caused by gentamicin administration. These findings highlight potential avenues for the development of therapeutic strategies against gentamicin-induced renal disturbances.

ALA inhibiting the RRK pathway could offer various benefits to renal function and structure. The RRK pathway mediates renal vasculature vasoconstriction, particularly in the afferent and efferent arterioles (*Guan et al., 2019*). ALA may improve the renal blood flow and glomerular filtration rate by blocking this mechanism. The RRK pathway regulates the actin cytoskeleton and cell-cell and cell-matrix interactions in various renal cells, including tubular epithelial cells, mesangial cells, and podocytes (*Jiang, Sha & Schacht,*

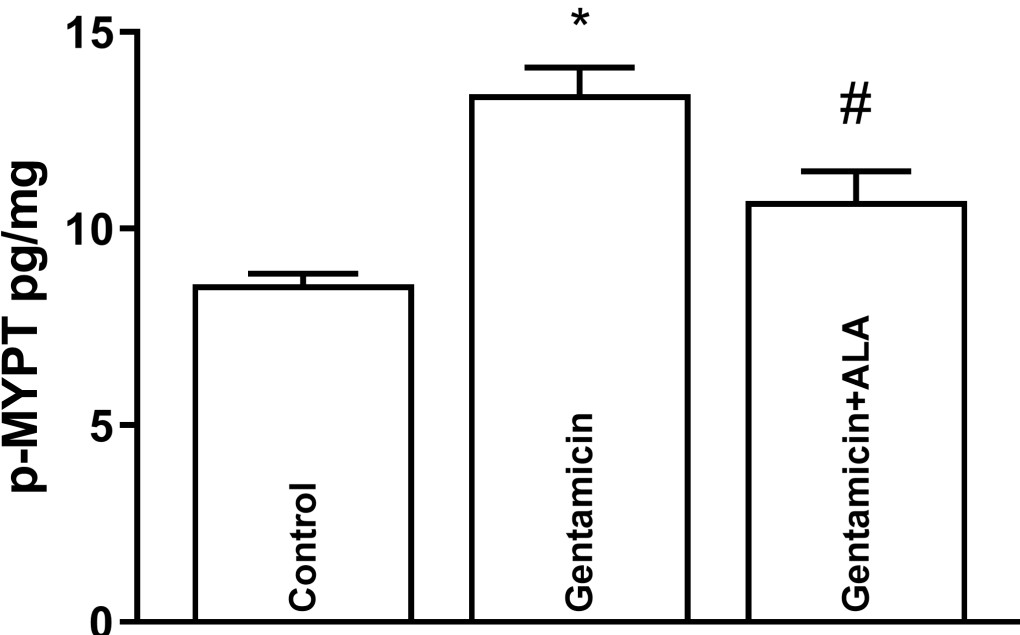

**Figure 3** **The impact of gentamicin and ALA treatments on Rho-kinase activity (measured by ELISA) in the kidneys.** *: $p < 0.05$ when compared to control, #: $p < 0.05$ when compared to gentamicin treated group, $n = 8$.

**Table 1** **Mean values of Rho-A, Rho-kinase II concentrations and Rho-kinase activity for two control, gentamicin and gentamicin+ALA treated groups ($n = 8$ for each group).**

|  | Control group (S.E.M) | Gentamicin group (S.E.M) ($p$-value) | Gentamicin + ALA group (S.E.M) ($p$-value) |
|---|---|---|---|
| Rho-A pg/ml | 1,260 (166.50) | 1,755 (131.20) (0.0349) | 1,119 (67.08) (0.0053) |
| Rho-kinase II pg/ml | 32,265 (1,938) | 47,218 (977) ($p < 0.0001$) | 39,808 (858) ($p < 0.0001$) |
| Rho-kinase activity p-MYPT pg/mg | 8.576 (0.2810) | 13.42 (0.6826) ($p < 0.0001$) | 10.71 (0.7466) ($p < 0.0001$) |

*2006*). By inhibiting this pathway, ALA may prevent gentamicin-induced loss of the brush border, the mesangial expansion, and the podocyte injury. Moreover, the RRK pathway is associated with the activation of several transcription factors and the expression of pro-inflammatory and pro-fibrotic mediators, such as nuclear factor-kappa B, tumor necrosis factor-alpha, monocyte chemoattractant protein-1, and connective tissue growth

factor (*Hayashi et al., 2006*; *Jiang et al., 2016*; *Ruperez et al., 2005*). ALA restricting this pathway may reduce the inflammation and fibrosis induced by gentamicin.

The elevation in Rho-kinase activity after gentamicin administration adds complexity to the study. Rho-kinase activation exacerbates renal vasoconstriction and inflammation, contributing to renal injury (*Cavarape et al., 2003*). The effective reversal of this increase by ALA suggests a potential mechanism for its protective effects. This antioxidant may help alleviate the vasoconstriction and inflammation associated with gentamicin treatment by downregulating Rho-kinase activity.

## CONCLUSIONS

In conclusion, this study demonstrates that oxidative stress plays a role in the activation of the RRK pathway in gentamicin-induced renal injury (*Seccia et al., 2020*; *Su et al., 2021*). ALA inhibits the RRK pathway and may protect kidneys from damage caused by gentamicin (*Dik, Hatipoglu & Ates, 2024*; *Priyadarshini, Aatif & Bano, 2012*). These findings suggest that the modulation of the RRK pathway by antioxidants may be a novel therapeutic strategy for the prevention and treatment of gentamicin-induced nephrotoxicity.

Study limitations encompass the brief treatment duration, absence of histological and functional renal injury assessment, and the utilization of a single dose for gentamicin and ALA. Future studies should extend the treatment duration, evaluate histological and functional kidney changes, and optimize the dose and timing of gentamicin and ALA administration. Additionally, the molecular mechanisms responsible for the antioxidant effects of ALA on the RRK pathway should be further explored.

### Funding
The authors received no funding for this work.

### Competing Interests
The authors declare there are no competing interests.

### Author Contributions
- Percin Pazarci conceived and designed the experiments, performed the experiments, analyzed the data, prepared figures and/or tables, authored or reviewed drafts of the article, and approved the final draft.
- Serkan Özler conceived and designed the experiments, performed the experiments, analyzed the data, prepared figures and/or tables, authored or reviewed drafts of the article, and approved the final draft.
- Halil Mahir Kaplan conceived and designed the experiments, performed the experiments, analyzed the data, prepared figures and/or tables, authored or reviewed drafts of the article, and approved the final draft.

## Animal Ethics

The following information was supplied relating to ethical approvals (i.e., approving body and any reference numbers):

Cukurova University Animal Care and Ethics Committee (protocol code 2016-2-13)

## Data Availability

The concentrations measured by ELISA method (standardized with Bradford method) for proteins RhoA rho-kinase II and Rho-kinase activity are available in the Supplemental Files.

## Supplemental Information

Supplemental information for this article can be found online at http://dx.doi.org/10.7717/peerj.18335#supplemental-information.

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
