# Peer review of "Effect of alpha-linolenic acid on aminoglycoside nephrotoxicity and RhoA/Rho-kinase pathway in kidney"

_PeerJ, doi:10.7717/peerj.18335_

## Round 0.1 · original submission · Major Revisions

Please address all the concerns of the reviewers and make the required changes in the manuscript.

·

Basic reporting

This paper is well-written and includes a relevant introduction and background addressed in simple terms which makes comprehension easy. However, some aspect requires proper citation, notably the claims in lines 41-44 and 117-120.

Experimental design

More information is required for the source of the reagents and instrumentation used to aid replication.
Also, the author mentioned carrying out protein quantification using the Bradford method but there is no area in the manuscripts on the result of the analysis.

Validity of the findings

The data provided is in-line with the research aims and conclusion, however, more references should be made to similar work in literature in the conclusion. Also, the method used in the work is ELISA, it should be included in the Figure description text.

Reviewer 2 ·

Basic reporting

In this manuscript, Pazarci et al. explore the effect of gentamicin and alpha-linolenic acid (ALA) on the levels of RhoA and rho-kinase II. While the topic is relevant and addresses an important area of research, the manuscript requires substantial revisions to meet the standards for publication in PeerJ. Below, I outline several key concerns that should be addressed before reconsideration.

1. Scientific Language: The manuscript lacks the precision and formality expected in scientific writing. The current narrative style detracts from the clarity and objectivity essential for effective scientific communication. A thorough revision is necessary to align the language with the standards of PeerJ, ensuring that it is both precise and formal.

2. Insufficient Data: The data presented are not sufficient to fully support the conclusions drawn. To enhance the robustness of the study, the inclusion of more comprehensive data is essential. This should involve additional experimental replicates and thorough statistical analyses to ensure the findings are both reliable and reproducible. For instance, only a single concentration of ALA (70 mg/kg) was tested, and the duration of the mouse study was limited to 9 days. It would be beneficial for the authors to provide a rationale for these specific conditions. Moreover, additional data points, including varying ALA concentrations and extended time points, are necessary to robustly support the conclusions of this work.

3. Missing Raw Data: The manuscript does not provide the raw data associated with the experiments. Transparency and reproducibility are crucial in scientific research, and the inclusion of raw data is necessary to uphold these principles. I recommend that the authors include the raw data either within the manuscript or as supplementary material.

4. Incomplete ELISA Experimental Details: The details regarding the key and only ELISA experiments are incomplete. The manuscript should specify the catalog and lot numbers for the kits used, as well as the instrument used for the measurements. This information is critical for the reproducibility of the experiments and should be clearly documented. The reliance solely on ELISA data to suggest differences in RhoA and rho-kinase II levels is insufficient to substantiate the authors' claims. To provide a more comprehensive analysis, additional supporting experiments, such as Western blotting and LC-MS/MS, are necessary. These experiments would help validate the findings and offer a deeper insight into the molecular mechanisms involved.

Experimental design

no comment

Validity of the findings

no comment

Additional comments

no comment

·

Basic reporting

1. In lines 26-27, please include the dosage of gentamicin and ALA used in the abstract.
2. In lines 45-52, including how ROCK1 & ROCK2 are related to tubular necrosis.
3. Include more background literature on how gentamicin affects renal necrosis in lines 58-64
4. Include a small summary of how ALA's antioxidant properties reduce oxidative stress in the Introduction.
5. In lines 102-104, include the catalog# of the ELISA kit used in the analysis and a brief description of the ELISA procedure used in the experiment.
6. In line 107, please specify the type of ANOVA—is it a one-way or two-way ANOVA used for the analysis? Also, mention any post hoc tests used in the experiment.
7. In lines 111-115, please include the p-value in the results section for the treatments.

Experimental design

1. Explain why you chose 100mg/kg doses of gentamicin and ALA for the treatment, any particular reason in lines 74-75.
2. Why did you include both gentamicin and ALA as one treatment group, are there any interactions between them?

Validity of the findings

1. In lines 123-124, you mentioned the chronic application of gentamicin, what is the exact meaning of chronic application?
2. I strongly suggest to rewrite the sentences in lines 125-132 for a better understanding of the discussion section.

---

## Round 0.2 · accepted · Accept

All concerns of the reviewers were adequately addressed, and the revised manuscript is acceptable now.

·

Basic reporting

All the initial comments and suggestions in the first review have been fully addressed.

Experimental design

All the initial comments and suggestions in the first review have been fully addressed.

Validity of the findings

All the initial comments and suggestions in the first review have been fully addressed.

Additional comments

All the initial comments and suggestions in the first review have been fully addressed.

Reviewer 2 ·

Basic reporting

The authors addressed all my concerns in the revision and the current version of this manuscript is suitable for publication.

Experimental design

no comment

Validity of the findings

no comment

·

Basic reporting

no comment

Experimental design

no comment

Validity of the findings

no comment